# Scalable Zeroth-Order Fine-Tuning for Extremely Large Language Models with Limited GPU Memory

**Liangyu Wang**[1], **Jie Ren**[1], **Hang Xu**[1], **Junxiao Wang**[2], **Huanyi Xie**[1],
**David E. Keyes**[1], **Di Wang**[1]

[1]King Abdullah University of Science and Technology (KAUST)
[2]Guangzhou University

{liangyu.wang}@kaust.edu.sa

## Abstract

Fine-tuning large pre-trained LLMs generally demands extensive GPU memory. Traditional first-order optimizers like SGD encounter substantial difficulties due to increased memory requirements from storing activations and gradients during both the forward and backward phases as the model size expands. Alternatively, zeroth-order (ZO) techniques can compute gradients using just forward operations, eliminating the need to store activations. Furthermore, by leveraging CPU capabilities, it's feasible to enhance both the memory and processing power available to a single GPU. We propose a novel framework, ZO2 (Zeroth-Order Offloading), for efficient zeroth-order fine-tuning of LLMs with only limited GPU memory. Our framework dynamically shifts model parameters between the CPU and GPU as required, optimizing computation flow and maximizing GPU usage by minimizing downtime. This integration of parameter adjustments with ZO's double forward operations reduces unnecessary data movement, enhancing the fine-tuning efficacy. Additionally, our framework supports an innovative low-bit precision approach in AMP (Automatic Mixed Precision) mode to streamline data exchanges between the CPU and GPU. Employing this approach allows us to fine-tune extraordinarily large models, such as the OPT-175B with 175 billion parameters, on a mere 18GB GPU. Moreover, our framework achieves these results with almost no additional time overhead and absolutely no accuracy loss compared to standard zeroth-order methods. ZO2's code has been open-sourced in *https://github.com/liangyuwang/zo2*.

## 1 Introduction

As Large Language Models (LLMs) grow in scale—reaching hundreds of billions of parameters (like OPT-175B (Zhang et al., 2022), Llama 3.1 405B (Dubey et al., 2024))—and their applications diversify (Pei et al., 2024; Huang et al., 2025; Pei et al., 2025), managing GPU memory resources effectively becomes crucial. Efficient GPU memory management is crucial not only because it directly influences model performance and training speed, but also because GPU memory is both expensive and limited in quantity. However, this creates a significant challenge in handling ever-larger models within the physical constraints of current hardware technologies. CPU offloading has become a crucial technique for overcoming the challenge. It involves transferring computations and data from the GPU to the CPU, specifically targeting data or parameters that are less frequently accessed ("inactive"). Specifically, it leverages the typically larger and more cost-effective CPU memory (DDR SDRAM) compared to the more expensive and less abundant GPU memory (HBM). By offloading these inactive tensors of the neural network, CPU offloading effectively alleviates the memory and computational pressures on GPUs. While CPU offloading has been commonly applied in inference to manage memory-intensive tasks like KV cache offloading (Ge

et al., 2023; Sheng et al., 2023) and Mixture of Experts (MoE) offloading (Eliseev & Mazur, 2023; Xue et al., 2024), its application in training, especially fine-tuning (Ouyang et al., 2022; Hu et al., 2022; Shao et al., 2024; Zheng et al., 2025; Wang et al., 2025), remains less explored.

Recently, some works (Rajbhandari et al., 2020; Ren et al., 2021) have tried to introduce CPU offloading into LLM training. However, they are typically constrained by the capabilities of first-order optimizers such as SGD and Adaptive Moment Estimation (AdamW) (Loshchilov & Hutter, 2017), and limited GPU memory, restricting large-scale model scalability on single GPU systems. In detail, using first-order optimizers introduces two major inefficiencies in CPU offloading (Section 4.1): **(1) Multiple communication operations**: During the training of LLMs, parameters are used not only for computing the loss during the forward pass but also for gradient computation in the backward pass. This necessitates offloading the same data (parameter) twice—once for each pass (see Figure 2a for an illustration). Such redundancy not only doubles the communication volume between the CPU and GPU but also introduces significant latency and inefficiency due to repetitive data transfers. **(2) Huge data transfer volume per communication operation**: Furthermore, both parameters and activations (hidden states) are required in the backward pass to complete gradient computations. This means that parameters and activation values must be offloaded during each forward pass and re-uploaded to the GPU for the backward pass. The result is a significant increase in the volume of data transferred, which severely impacts training throughput and efficiency.

On the other hand, compared to first-order optimization methods, zeroth-order (ZO) methods offer a novel approach to fine-tuning LLMs (Zhang et al., 2024; Malladi et al., 2023; Gautam et al., 2024). These methods utilize dual-forward passes to estimate parameter gradients and subsequently update parameters, as illustrated in Figure 2b. This approach eliminates the traditional reliance on backward passes, thereby streamlining the training process by significantly reducing the number of computational steps required.

Based on the above observations, we conjecture that ZO's architecture is particularly well-suited for CPU offloading strategies. Intuitively, by eliminating backward passes and the need to store activation values, it can significantly reduce GPU memory demands through efficient parameter offloading. However, despite these advantages, ZO training via CPU offloading introduces new challenges, particularly in the realm of CPU-to-GPU communication.

One challenge lies in ensuring the **alignment of random perturbations** used in dual-forward passes across different transformer blocks. In contrast to the monolithic forward pass in MeZO, our block-wise execution requires careful management of the random number generator (RNG) state to preserve gradient correctness. Without this mechanism, ZO training may suffer from accuracy mismatch.

Another key challenge is designing an efficient communication-computation scheduling mechanism. First-order offloading systems (Ren et al., 2021; Rajbhandari et al., 2021) and inference pipelines (Sheng et al., 2023) require intricate and often brittle scheduler logic to pipeline forward/backward steps with parameter movement. In contrast, we find that the **dual-forward nature of ZO optimization naturally doubles computation time**, enabling the communication to be fully hidden behind computation in most cases. This allows our dynamic scheduler to be **extremely simple yet highly effective**, requiring only three CUDA streams with minimal control logic.

To tackle these challenges, we introduce **ZO2 (Zeroth-Order Offloading)**, a novel framework specifically designed for zeroth-order fine-tuning of LLMs under tight memory constraints. Our design (1) preserves the mathematical integrity of MeZO by introducing a robust RNG state manager, and (2) achieves high throughput via a lightweight dynamic scheduler that overlaps parameter uploads, dual-forward computations, and offloads. In addition to the scheduler and RNG state management, ZO2 incorporates several complementary system optimizations to further enhance throughput and memory efficiency. These include a reusable memory strategy that eliminates repeated CUDA allocations, a reordering of parameter update operations to minimize data transfers, and AMP-mode-aware parameter

compression that reduces communication overhead during mixed-precision training. Our contributions can be summarized as follows:

- **Innovative use of CPU-offloading for ZO methods**: We determine that ZO is inherently more suitable for CPU offloading than first-order optimizers, due to its forward-only execution and minimal activation dependencies. By combining ZO with CPU-offloading, ZO2 enables efficient training of models like OPT-175B on a single 18GB GPU.

- **Completely lossless accuracy, high-throughput core design**: **(1)** We introduce an RNG state manager to ensure accuracy alignment between perturbation and parameter update phases during block-wise ZO training. **(2)** We propose a lightweight dynamic scheduler that overlaps parameter uploading, dual-forward computation, and offloading using only three CUDA streams. Compared to other complex scheduler logic design, ZO2 achieves simpler and more robust execution with minimal scheduling complexity.

- **Complementary system-level optimizations**: ZO2 integrates additional key improvements to maximize throughput and minimize overhead: **(1) Reusable memory allocation** to avoid costly CUDA malloc/free during block transfers (Section D.2). **(2) Efficient parameter update reordering** to reduce redundant data transfers by fusing updates into the forward phase (Section D.3). **(3) AMP-mode-aware compression** that reduces CPU-GPU communication volume via low-bit representations like FP16 and FP8 (Section D.4).

- **Empirical validation at scale**: ZO2 is able to fine-tune OPT-175B using just 18GB of GPU memory, with no additional runtime or accuracy degradation. Our ablation study confirms the contribution of each optimization component to overall throughput.

- **Open-source implementation with a clean and extensible API.** We release the full implementation of ZO2 with a lightweight and modular codebase. Our API (Appendix E) is intentionally designed to be simple and minimal, enabling users to easily plug ZO2 into existing PyTorch training workflows.

## 2 Related Work

**Zeroth-Order (ZO) Optimization.** ZO optimization offers a gradient-free alternative to first-order (FO) optimization by approximating gradients through function value-based estimates. These estimates theoretically require only two forward passes but are believed to be prohibitively slow for optimizing large models. Despite this limitation, ZO methods have been utilized in deep learning to generate adversarial examples or adjust input embeddings (Sun et al., 2022a;b), though they have not been widely adopted for direct optimization of large-scale models (Liu et al., 2020). Several acceleration techniques have been proposed to address the scaling challenges of ZO optimization and some of them have been used for LLM fine-tuning. These include using historical data to improve gradient estimators (Cheng et al., 2021), exploiting gradient structures (Singhal et al., 2023) or sparsity to reduce the dependence of ZO methods on the size of the problem (Chen et al., 2024; Cai et al., 2022; 2021), and reusing intermediate features (Chen et al., 2024) and random perturbation vectors (Malladi et al., 2023) during the optimization process. These advancements suggest that ZO optimization could increasingly be applied to more complex and large-scale ML problems. While previous ZO optimization efforts have primarily targeted algorithmic improvements for GPU memory efficiency, our approach extends these optimizations to the system level, enabling more robust memory management and enhanced performance for large-scale machine learning applications. *More related work can be found in Appendix B.*

## 3 Preliminaries on Zeroth-Order Optimization

Zeroth-order (ZO) optimization provides a gradient-free alternative to first-order (FO) methods, estimating gradients using only function evaluations. In this paper, we focus on

the **Randomized Gradient Estimator (RGE)** (Nesterov & Spokoiny, 2017), which forms the foundation of MeZO (Malladi et al., 2023) and our work.

RGE approximates the gradient $\nabla L(\theta)$ by evaluating the loss function along a randomly sampled direction $z \sim \mathcal{N}(0, I)$. The projected gradient is defined as:

$$g = \frac{L(\theta + \epsilon z) - L(\theta - \epsilon z)}{2\epsilon} \in \mathbb{R}^1, \tag{1}$$

and the parameter update is given by:

$$\theta \leftarrow \theta - \eta \cdot g \cdot z, \tag{2}$$

where $\epsilon$ is a small smoothing parameter and $\eta$ is the learning rate.

Importantly, ZO methods avoid the need for backward computation and activation storage by relying solely on dual-forward passes. This forward-only nature, combined with the fact that $g$ is a scalar, makes ZO highly memory-efficient and well-suited for offloading-based training. *For a full derivation and the complete MeZO algorithm, please refer to Appendix C.*

## 4 Framework Overview

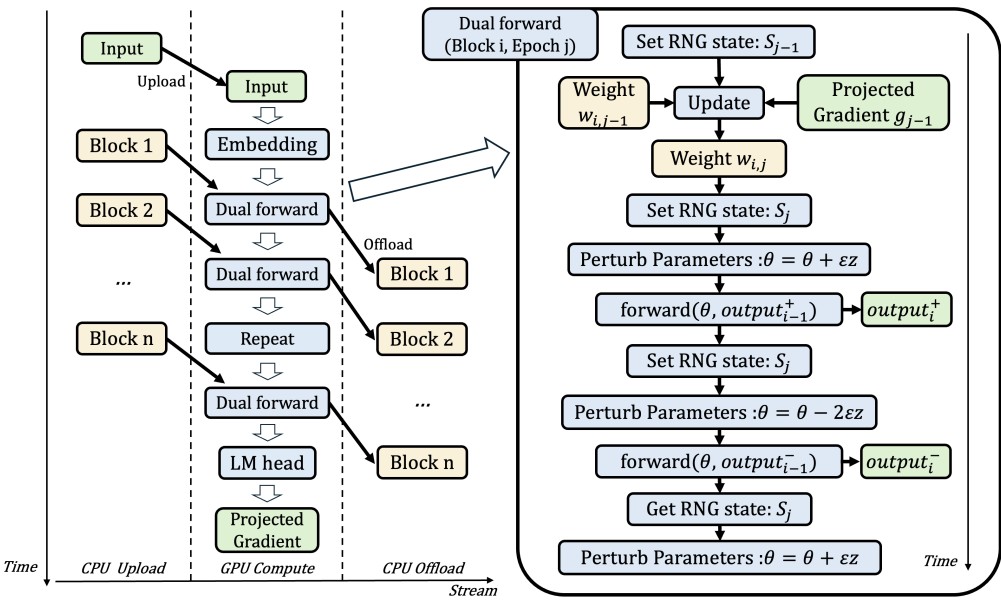

Figure 1: Workflow of the ZO2 framework for fine-tuning LLMs.

### 4.1 Insight

Our framework, ZO2, exclusively designed for zeroth-order optimizer, incorporates a CPU offloading strategy optimized for this specific approach. This design leverages the CPU for storage and the GPU for computation, streamlining parameter offloading in a manner uniquely suited to the operational characteristics of zeroth-order methods:

**(1)** As illustrated in Figure 2(a), the first-order optimizer typically requires both forward and backward passes with opposing workflow directions, and each pass necessitates parameter availability. Consequently, when parameters offloaded to the CPU are required for computation on the GPU, communication is necessary twice (forward and backward passes). In contrast, the zeroth-order optimizer necessitates only two forward passes with the same workflow direction (Figure 2(b)), allowing parameters to be reused with just one communication cycle. This modification effectively halves the communication frequency.

(a) Model using first-order optimizer with forward-backward passes workflow

(b) Model using zero-order optimizer with only dual-forward passes workflow

Figure 2: **Motivation.** Using linear operator as an example. **(a) First-Order Optimizer:** Requires forward and backward passes, storing intermediate activations $(X_1, X_2)$ and incurring repeated CPU-GPU parameter transfers. **(b) Zeroth-Order Optimizer:** Uses dual-forward passes with perturbed weights $(W_i, W_i')$ to estimate gradients. No activation storage is needed, and each parameter is transferred only once, reducing memory and communication overhead.

**(2)** As discussed in Section 5.2, it is essential to overlap communication and computation tasks. Normally, communication between the CPU and GPU consumes significantly more time than GPU computations. The zeroth-order optimizer, with its dual-forward passes in the same direction, maintains the duration of communication while increasing computation time. This setup effectively reduces the overall communication overhead by half and reduce the complexity of the scheduler.

**(3)** The zeroth-order optimizer does not involve a backward pass, eliminating the need to store and offload activations. This feature significantly reduces memory requirements.

**(4)** The gradient calculation in the zeroth-order optimizer is achieved by multiplying a projected gradient with a Gaussian distribution. The projected gradient is a single value (Equation 1), and the Gaussian distribution can be generated using a seed. Consequently, the actual gradient can be computed in-place during parameter updates, eliminating the need for dedicated storage space for real gradients.

### 4.2 ZO2 Framework Overview

In this part, we first provide an overview and a brief introduction to our ZO2 (Zeroth-Order Offloading) framework. To better illustrate our idea, we first describe the computation workflow of the original ZO optimization procedure for LLM fine-tuning. Consider the architecture of a simple decoder-only LLM, which typically comprises an embedding layer, $N$ transformer blocks, and a language model (LM) head layer. Our offloading strategy involves offloading all transformer blocks to the CPU, while retaining the remaining components on the GPU. This approach is similarly applicable to more complex LLM architectures like OPT (Zhang et al., 2022). Initially, input data is loaded from the disk into the CPU and subsequently transferred to the GPU. Within the GPU, each module—including the embedding layer, transformer blocks, and the language model (LM) head—executes dual-forward computations sequentially to estimate the projected gradient and update parameters.

The above approach divides the entire model's dual-forward process, as outlined in the original ZO workflow (Algorithm 3), into discrete block-level operations. However, this could potentially introduce an accuracy mismatch. This challenge arises because the core of the ZO method relies on the uniformity of Gaussian random vectors applied during both the perturbation and the parameter update phases, which must be consistently synchronized across all computations (Algorithm 3). To address this, we propose a random number generator (RNG) state manager (Section 5.1) that meticulously aligns the random vectors. This management ensures that the random perturbations and the subsequent parameter updates across different transformer blocks maintain identical stochastic characteristics.

From the efficiency perspective, naive implementation with deep learning frameworks like PyTorch (Paszke et al., 2019) typically manage both communication (via interconnections, e.g., PCIe) and computation tasks with a single CUDA stream, leading to significant inefficiencies. Specifically, for ZO optimization, the $i$-th transformer block is uploaded from the CPU to the GPU (the GPU is designated for computation-intensive tasks using its CUDA and Tensor Cores, and the CPU memory is used for parameter storage), undergoes dual-forward

computation, and then is offloaded back to the CPU. The $i + 1$-th block must wait for the offloading of the $i$-th block to finish before its uploading, leading to idle CUDA and Tensor Cores during communication while the interconnection remains idle during computation. See Figure 5 for an illustration. Our ZO2 framework implementation achieves the strategic utilization of CPU and GPU resources (Section 5.2). This approach involves dynamically offloading model parameters to the CPU and uploading them back to the GPU as needed for computation. Specifically, for the transformer model structure, each transformer block is individually uploaded for processing and subsequently offloaded post-computation, thus balancing communication and computation across blocks. As illustrated in Figure 1, while the $i$-th transformer block is being computed, the $i + 1$-th block is pre-uploaded, and the $i - 1$-th block is offloaded simultaneously. This strategic overlapping ensures continuous and efficient computation, reducing idle times and maximizing GPU utilization. In the uploading phase of ZO2, transformer blocks are transferred into a reusable memory space on the GPU, eliminating the extra time typically required for CUDA memory allocation (Section D.2). Moreover, parameter updates are ingeniously fused with the dual-forward passes to minimize redundant data transfers, thereby enhancing the overall efficiency of the model training process (Section D.3).

---

**Algorithm 1** ZO2 Computation with RNG State Manager

---

**Require:** Transformer blocks $\{W_i\}_{i=1}^{N}$ with number of transformer blocks $N$, embedding parameters *Embedding*, and LM head *LMhead*, module parameter $\theta$, module forward function *forward*, loss function $L$, training iterations $T$, perturbation step size $\epsilon$, data batch $B$, learning rate $\eta_t$, random seed $s$, random state buffer $rsb$, last iteration's random state $lrs$, function UpdateParameters and PerturbParameters from Algorithm 3.
1: Initialize $g = 0$.
2: **for** $j = 1, \ldots, T$ **do**
3:     Set random seed $s$ and sample batch $B \subset D$
4:     Get random state $rs$ = GetRngState($s$), and push $rs$ into $rsb$.
5:     **if** $j > 1$ **then**
6:         Update last iteration's random state $lrs$ = PopLeft($rsb$)
7:     **else**
8:         $lrs$ = *None*
9:     **end if**
10:     $out_+, out_-, rs, lrs$ = DualForward($Embedding, \epsilon, s, rs, lrs, g, B, B$)
11:     **for** $i = 1$ to $N$ **do**
12:         $out_+, out_-, rs, lrs$ = DualForward($W_i, \epsilon, s, rs, lrs, g, out_+, out_-$)
13:     **end for**
14:     $out_+, out_-, rs, lrs$ = DualForward($LMhead, \epsilon, s, rs, lrs, g, out_+, out_-$)
15:     $\ell_+ = L(out_+), \ell_- = L(out_-)$
16:     $g \leftarrow (\ell_+ - \ell_-)/(2\epsilon)$
17: **end for**

18: **function** DUALFORWARD($\theta, \epsilon, s, rs, lrs, g, input_+, input_-$)
19:     **if** $g$ != 0 **then**
20:         SetRngState($s, lrs$)
21:         $\theta, lrs \leftarrow$ UpdateParameters($\theta, g$)
22:     **end if**
23:     SetRngState($s, rs$), $\theta \leftarrow$ PerturbParameters($\theta, \epsilon$)
24:     $out_+ \leftarrow forward(\theta; input_+)$
25:     SetRngState($s, rs$), $\theta \leftarrow$ PerturbParameters($\theta, -2\epsilon$)
26:     $out_- \leftarrow forward(\theta; input_-)$
27:     SetRngState($s, rs$), $\theta \leftarrow$ PerturbParameters($\theta, \epsilon$)
28:     $rs$ = GetRngState($s$)
29:     **return** $out_+, out_-, rs, lrs$
30: **end function**

---

Our ZO2 framework further integrates a novel low-bit precision technique that efficiently manages data transfers between the CPU and GPU in the AMP mode (see Figure 4 for

an illustration). This technique is aligned with AMP protocols by ensuring that high-bit precision is maintained for parameter updates, while low-bit precision data is used for computation on the GPU (Section D.4). This dual-precision approach significantly reduces the communication overhead, optimizing memory usage without compromising computational accuracy.

In the following section, we will provide implementation challenges and details of our framework.

## 5 Design and Implementation Details

### 5.1 ZO2 with RNG State Manager

The core principle of ZO algorithms is the uniform application of Gaussian random vector for each parameter during both the perturbation and update phases. MeZO (Algorithm 3) accomplishes this by resetting the seed at each iteration to control the state of the random number generator (RNG), ensuring consistent execution. However, unlike MeZO, ZO2 disaggregates the model's dual-forward process across different model blocks (Figure 1, Algorithm 1), which could potentially lead to discrepancies in the RNG states between perturbation and parameter updates.

To maintain the precision of the MeZO workflow within the ZO2 framework, we meticulously record the RNG states (*rng_state*) during each module's dual-forward operation (Algorithm 1 Line 18-30). Specifically, *rng_state* is saved prior to executing any parameter perturbations and before the module forward pass. This ensures that outputs are consistently generated across iterations. Additionally, given that parameters are updated using the gradient projected from the last iteration (refer to Section D.3), we preserve the last random state (*last_rstate*) to accurately replicate the Gaussian perturbations that were applied during the perturbation process (Algorithm 1 Line 4-9).

This precise synchronization of *rng_state* and *last_rstate* across different model blocks in ZO2 guarantees that each parameter update adheres to the same stochastic path as established in MeZO. Consequently, this methodological rigor ensures the preservation of exact accuracy throughout the workflow, facilitating reliable and reproducible outcomes in line with the original algorithmic design.

### 5.2 Dynamic Scheduler Design for Efficient Overlap

---

**Algorithm 2** ZO2 Dynamic Scheduler

---

**Require:** Transformer blocks $\{W_i\}_{i=1}^N$ with number of transformer blocks $N$, embedding parameters *Embedding*, and LM head *LMhead*.
 1: Initialize a dynamic scheduler $S\{\cdot\}$ to control dual-forward computation $C(\cdot)$, uploading $U(\cdot)$, and offloading $O(\cdot)$ operations.
 2: Asynchronously launch $S\{C(Embedding), U(W_1)\}$.
 3: **for** $i = 1$ to $N - 1$ **do**
 4:     Synchronously wait until $U(W_i)$ finished.
 5:     **if** $i = 1$ **then**
 6:         Asynchronously launch $S\{C(W_i), U(W_{i+1})\}$.
 7:     **else**
 8:         Synchronously wait until $C(W_{i-1})$ finished.
 9:         Asynchronously launch $S\{O(W_{i-1}), C(W_i), U(W_{i+1})\}$.
10:     **end if**
11: **end for**
12: Synchronously wait until $C(W_{N-1})$ and $U(W_N)$ finished.
13: Asynchronously launch $S\{O(W_{N-1}), C(W_N)\}$.
14: Synchronously wait until $C(W_N)$ finished.
15: Asynchronously launch $S\{O(W_N), C(LMhead)\}$.

---

While Section 5.1 ensures the *correctness and precision alignment* of zeroth-order updates by managing random number generator (RNG) states within the `DualForward` routine (Algorithm 1), this section focuses on improving *efficiency* by overlapping CPU-GPU communication with computation. Specifically, we aim to accelerate the core task in Algorithm 1—the `DualForward` computation for each transformer block—by overlapping it with parameter uploads and offloads.

Figure 5 illustrates the evolution from a *naive sequential execution* (Figure 5a), where all communication and computation are serialized, to an *overlapped execution model* (Figure 5b), which forms the basis of our dynamic scheduler. In this optimized setting, computation of block $i$ (`DualForward($W_i$)`), uploading of block $i{+}1$, and offloading of block $i{-}1$ are launched concurrently on **three dedicated CUDA streams**.

A key enabler of this simplified scheduling design is the **dual-forward nature of ZO optimization**, which inherently doubles the computation time while leaving the communication time per block unchanged. This *high compute-to-communication ratio* makes it much easier to achieve full overlap, effectively hiding communication delays. This behavior stands in sharp contrast with traditional first-order offloading training methods (Ren et al., 2021; Rajbhandari et al., 2021), where scheduling must delicately interleave forward, backward, and optimizer operations. Similarly, inference frameworks like (Sheng et al., 2023) require complex runtime stream orchestration. In contrast, ZO2 achieves efficient scheduling with *minimal control logic*.

To ensure correctness during asynchronous execution, we enforce two dependency rules: **(1) Intra-block**: offload of block $i$ waits for completion of compute, which waits for upload. **(2) Inter-block**: each upload/compute/offload task waits for the previous block's corresponding task. Our actual execution pattern follows the sequence $O(W_{i-1}) \rightarrow C(W_i) \rightarrow U(W_{i+1})$, as formalized in Algorithm 2. This pattern ensures safety without requiring global synchronization, thus avoiding pipeline "bubbles".

We also retain the *embedding layer and LM head* on the GPU throughout. This not only avoids unnecessary transfers but also enables uploading of the first transformer block while the embedding layer is being computed—further improving overlap efficiency. *The full scheduler design section can be seen in Appendix D.1.*

## 6 Experiment

We evaluate ZO2 using the OPT model family (Zhang et al., 2022), ranging from 1.3B to 175B parameters (Table 4), to assess scalability across different model sizes. Our baseline is MeZO (Malladi et al., 2023), the most memory- and throughput-efficient zeroth-order optimizer to date. ZO2 builds upon MeZO, aiming to further reduce GPU memory usage without sacrificing throughput or accuracy. *More experimental setups and additional experiments can be found in Appendix F, G.*

### 6.1 Main Results

Table 1: **Main results of ZO2 performance for various model configurations and both FP32 and FP16 modes.** Instances of '-' in the table indicate scenarios where the corresponding method failed to execute due to memory constraints. The values in parentheses (x) represent the ratio of each measurement compared to the baseline MeZO (first column) configuration.

| Model | GPU Memory Usage (MB) ↓ | | | | Throughput (tokens/sec) ↑ | | | |
|---|---|---|---|---|---|---|---|---|
| | MeZO (FP32) | ZO2 (FP32) | MeZO (FP16) | ZO2 (FP16) | MeZO (FP32) | ZO2 (FP32) | MeZO (FP16) | ZO2 (FP16) |
| OPT-1.3B | 8898 | 5098(x0.57) | 5814(x0.65) | **3750(x0.42)** | 1998 | 1955(x0.97) | **6629(x3.32)** | 6448(x3.23) |
| OPT-2.7B | 14514 | 5930(x0.41) | 9054(x0.62) | **4142(x0.29)** | 1104 | 1086(x0.98) | **4229(x3.83)** | 4220(x3.82) |
| OPT-6.7B | 32930 | 8420(x0.26) | 16586(x0.50) | **4992(x0.15)** | 492 | 485(x0.98) | **2349(x4.77)** | 2270(x4.61) |
| OPT-13B | 58762 | 10736(x0.18) | 29690(x0.51) | **6180(x0.11)** | 266 | 259(x0.97) | **1326(x5.87)** | 1251(x5.54) |
| OPT-30B | - | 15981 | 63896 | **8856** | - | 122 | **641** | 514 |
| OPT-66B | - | 22295 | - | **12071** | - | 40 | - | **273** |
| OPT-175B | - | 34015 | - | **18039** | - | 14 | - | **37** |

The performance results of our experiments are presented in Table 1, where we compare the GPU memory usage and throughput of the MeZO and ZO2 frameworks, employing both FP32 and FP16 data formats. The results demonstrate a consistent advantage of ZO2 in terms of GPU memory utilization across all model sizes, highlighting significant efficiency improvements, especially in large-scale models like **OPT-175B**. This efficiency is attributed to ZO2's design, which strategically utilizes GPU memory to temporarily store only a limited number of transformer blocks for computation rather than the entire model. Notably, the memory savings become more pronounced as the model size increases. For smaller models, the GPU memory savings are less pronounced due to the significant proportion of memory allocated for input data, which diminishes the relative impact of the memory optimization.

In terms of throughput, ZO2 maintains a performance comparable to MeZO in most tested scenarios without any additional time overhead. The instances where ZO2 exhibits a decrease in throughput, such as with the OPT-1.3B model in FP32 format, can be primarily attributed to the dynamics of computation and communication. In these cases, the computation of each transformer block's dual-forward passes completes quicker than their corresponding communication tasks, leading to idle times as the dynamic scheduler (discussed in Section 5.2) synchronizes and waits for these communication tasks to conclude. It is important to note that our results do not show a consistent pattern where either smaller or larger models benefit more significantly from the computation-communication overlap, indicating that the effectiveness of this overlap does not linearly correlate with model size.

### 6.2 Results on Accuracy Alignment

Table 2: **Main results of ZO2 precision on OPT-13B**

| Method | SST-2 (%) | RTE (%) | CB (%) | BoolQ (%) | WSC (%) | WIC (%) | MultiRC (%) |
|--------|-----------|---------|--------|-----------|---------|---------|-------------|
| MeZO   | 91.4      | 66.1    | 67.9   | 67.6      | 63.5    | 61.1    | 60.1        |
| ZO2    | 91.4      | 66.1    | 67.9   | 67.6      | 63.5    | 61.1    | 60.1        |

We evaluate ZO2's accuracy alignment on the OPT-13B model across seven NLP benchmarks: SST-2, RTE, CB, BoolQ, WSC, WIC, and MultiRC (Socher et al., 2013; Dagan et al., 2005; De Marneffe et al., 2019; Clark et al., 2019; Levesque et al., 2012; Pilehvar & Camacho-Collados, 2018; Khashabi et al., 2018). These tasks span sentiment analysis, entailment, coreference, QA, and word sense disambiguation, covering diverse aspects of language understanding.

As shown in Table 2, ZO2 achieves identical precision rates to the baseline MeZO approach across all evaluated benchmarks. This parity in performance is significant as it not only validates the effectiveness of our RNG manager but also highlights ZO2's capability to maintain model precision while reducing GPU memory usage. Since ZO2 achieves the same accuracy as *MeZO*—which has already been extensively compared against FO methods in its original paper—we omitted redundant FO comparisons to keep the scope focused.

### 6.3 Ablation Study

Table 3: **Throughput (tokens/sec) results to validate proposed features.**

| Model | MeZO | ZO2 (no scheduler overlap) | ZO2 (no reusable memory) | ZO2 (no efficient update) | ZO2 |
|-------|------|----------------------------|--------------------------|---------------------------|-----|
| OPT-1.3B  | 1998 | 1109 (x0.56) | 735 (x0.37) | 1567 (x0.78) | 1955 (x0.97) |
| OPT-2.7B  | 1104 | 573 (x0.52)  | 422 (x0.38) | 849 (x0.77)  | 1086 (x0.98) |
| OPT-6.7B  | 492  | 225 (x0.46)  | 184 (x0.37) | 373 (x0.76)  | 485 (x0.98)  |
| OPT-13B   | 266  | 105 (x0.39)  | 103 (x0.39) | 198 (x0.74)  | 259 (x0.97)  |
| OPT-30B   | -    | 35           | 46          | 81           | 122          |
| OPT-66B   | -    | 22           | 15          | 36           | 40           |
| OPT-175B  | -    | 8            | 5           | 13           | 14           |

In order to discern the individual contributions of key features within the ZO2 framework to its overall performance, an ablation study was conducted focusing on three critical

components: the dynamic scheduler (Sec. 5.2), reusable memory (Sec. D.2), and efficient parameter updating (Sec. D.3). This study focused on throughput, as the three features were designed to improve it without affecting ZO2's built-in memory efficiency. Since CPU offloading already ensures low memory usage, an ablation on memory was unnecessary. Given the tightly integrated nature of our system, traditional ablation methodologies that add one feature at a time to a baseline are impractical. Instead, we adopted a reverse ablation approach where each feature was individually disabled. This allowed us to observe the decrement in throughput relative to the fully operational framework, thereby highlighting the significance of each component.

The results, presented in Table 3, provide a clear illustration of how the absence of each feature impacts the system's throughput: (1) **Horizontal Comparison.** Across all models, the removal of reusable memory results in the most substantial decrease in throughput, followed by the dynamic scheduler, and finally, the efficient parameter updating. This order of impact suggests that while all three features are pivotal, the overhead introduced by CUDA malloc operations, which are eliminated by reusable memory, significantly outweighs the communication delays between the CPU and GPU, managed by the dynamic scheduler and efficient parameter updating. For instance, when reusable memory is not employed, the throughput drops to 37% of the fully optimized framework for the OPT-6.7B model, highlighting its critical role in enhancing performance. (2) **Vertical Comparison.** As the model size increases, the relative importance of the dynamic scheduler and efficient parameter updating grows more pronounced. This trend is observable from the throughput: for larger models like OPT-6.7B, the reduction in throughput when the scheduler and efficient update features are disabled is relatively larger than in small models. This indicates that as models become larger, the complexities and overheads associated with managing and optimizing communications between CPU and GPU become more critical to maintaining performance.

## 7 Conclusion

In this paper, we presented ZO2, an efficient framework that enables the training of extremely large language models, such as the OPT-175B, with 18GB GPU memory—a capability previously unattainable with traditional methods. By effectively integrating CPU offloading, random number generator manager, high-performance dynamic scheduler, efficient memory management, efficient parameter updating, and AMP support, our framework reduces GPU memory demands while maintaining high throughput without additional time costs. These innovations not only lower the bar for teams with limited hardware resources and advance the democratization of large models, but also open new avenues for advancing AI technology more efficiently.

## Acknowledgment

Di Wang and Liangyu Wang are partially supported by KAUST through awards BAS/1/1689-01-01, URF/1/5508-01-01, and by the KAUST Center of Excellence for Generative AI under award number 5940.

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

# A   More Figures, Algorithms, and Tables

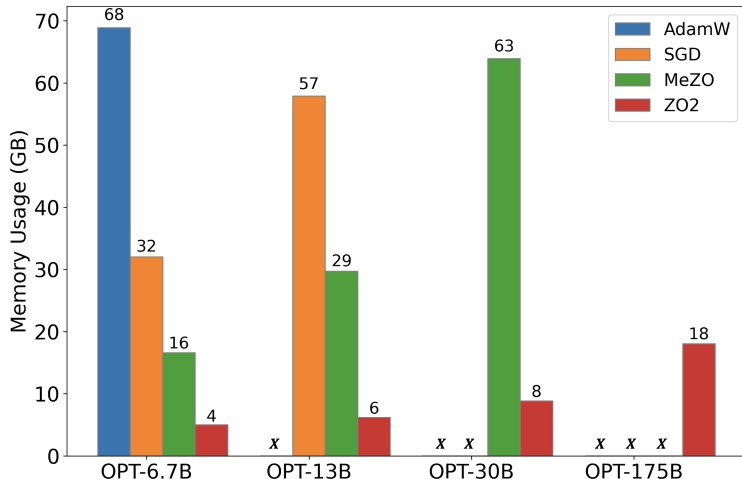

Figure 3: Single GPU memory usage comparison for training LLMs across different optimizers (AdamW, SGD, MeZO, and ZO2 (Zeroth-Order Offload)) and model sizes (OPT-6.7B, OPT-13B, OPT-30B, OPT-175B). The 'X' indicates that training was not feasible due to excessive memory demand.

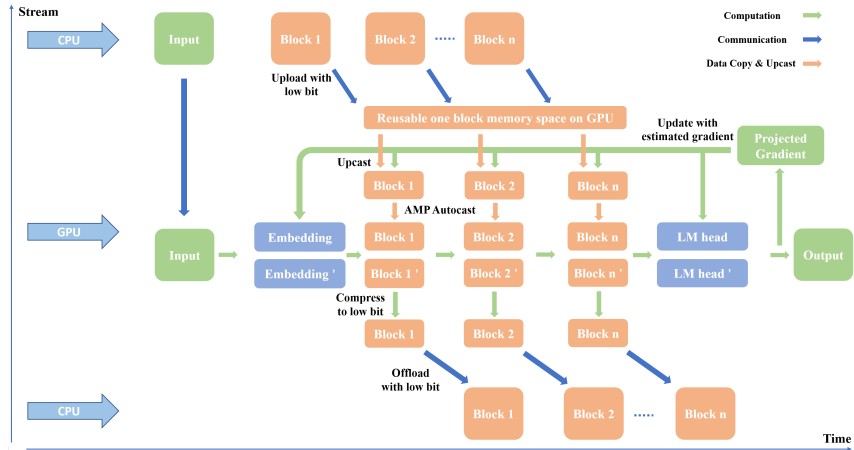

Figure 4: Workflow of the ZO2 framework (AMP mode) for fine-tuning LLMs.

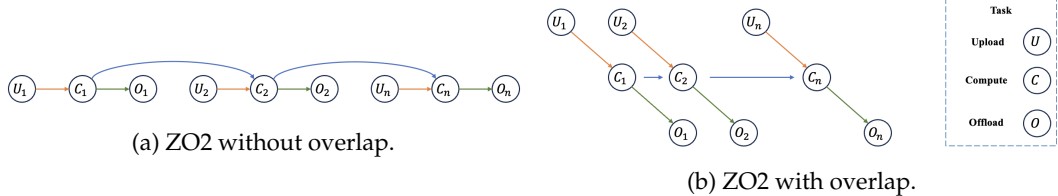

Figure 5: Sequential Task Execution in the Naive ZO2 Framework Depicting Non-overlapping Dual-Forward Passes and Associated Inefficiencies

---

**Algorithm 3** MeZO (Malladi et al., 2023)

---

**Require:** Model parameters $\theta \in \mathbb{R}^d$, loss function $L : \mathbb{R}^d \to \mathbb{R}$, training iterations $T$, perturbation step size $\epsilon$, data batch $B$, learning rate $\eta_t$
1: **for** $j = 1, \ldots, T$ **do**
2:     Set random seed $s$ and sample batch $B \subset D$
3:     $\theta \leftarrow \text{PerturbParameters}(\theta, \epsilon)$
4:     $\ell_+ \leftarrow L(\theta; B)$
5:     Reset RNG with seed $s$
6:     $\theta \leftarrow \text{PerturbParameters}(\theta, -2\epsilon)$
7:     $\ell_- \leftarrow L(\theta; B)$
8:     Reset RNG with seed $s$
9:     $\theta \leftarrow \text{PerturbParameters}(\theta, \epsilon)$
10:     $g \leftarrow (\ell_+ - \ell_-)/(2\epsilon)$
11:     Reset RNG with seed $s$
12:     $\theta \leftarrow \text{UpdateParameters}(\theta, g)$
13: **end for**

14: **function** UPDATEPARAMETERS($\theta, g$)
15:     **for** each $\theta_i \in \theta$, where $\theta_i \in \mathbb{R}^{d_i}$ **do**
16:         $z_i \sim \mathcal{N}(0, 1) \in \mathbb{R}^{d_i}$
17:         $\theta_i \leftarrow \theta_i - \eta_t \cdot g \cdot z_i$
18:     **end for**
19:     **return** $\theta$
20: **end function**

21: **function** PERTURBPARAMETERS($\theta, \epsilon$)
22:     **for** each $\theta_i \in \theta$, where $\theta_i \in \mathbb{R}^{d_i}$ **do**
23:         $z_i \sim \mathcal{N}(0, 1) \in \mathbb{R}^{d_i}$
24:         $\theta_i \leftarrow \theta_i + \epsilon z_i$
25:     **end for**
26:     **return** $\theta$
27: **end function**

---

Table 4: OPT model family configs in experiments.

| Model Size | Layers | Heads | Dimension | Sequence Length |
|---|---|---|---|---|
| 1.3B | 24 | 32 | 2048 | |
| 2.7B | 32 | 32 | 2560 | |
| 6.7B | 32 | 32 | 4096 | |
| 13B | 40 | 40 | 5120 | 2048 |
| 30B | 48 | 56 | 7168 | |
| 66B | 64 | 72 | 9216 | |
| 175B | 96 | 96 | 12288 | |

## B More Related Work

**CPU Offloading for LLMs.** With recent advancements in LLMs, several approaches have emerged to offload data to CPU memory, mitigating GPU memory limitations. One such method is vLLM (Kwon et al., 2023), which utilizes PagedAttention to dynamically manage the key-value (KV) cache at a granular block level. Portions of the KV cache can be temporarily swapped out of GPU memory to accommodate new requests. Llama.cpp (Gerganov, 2023) addresses oversized LLMs inference by using static layer partitioning. It stores certain contiguous layers in CPU memory while keeping others in GPU memory. During computation, the CPU handles the layers in its memory, followed by the GPU computing its assigned layers. FlexGen (Sheng et al., 2023), a GPU-centric inter-layer pipeline LLMs inference method, seeks to improve throughput by pinning some model weights in GPU memory for each layer. During inference, it overlaps GPU processing of the current layer with data loading for the next. DeepSpeed (Rajbhandari et al., 2020) introduces a technique to offload the first-order optimizer state to the CPU, significantly reducing GPU memory requirements during training. Zero-offload (Ren et al., 2021) extends the DeepSpeed approach by not only offloading data to the CPU but also engaging the CPU in computational tasks. Despite these advancements, the predominant focus of previous research has been on optimizing LLM inference or first-order optimization through strategic CPU-GPU data transfers. Our work, in contrast, introduces a novel approach by implementing CPU offloading specifically for zeroth-order optimization and fine-tuning of LLMs.

## C Full Preliminaries on ZO and ZO-SGD

ZO optimization offers a gradient-free alternative to first-order (FO) optimization by approximating gradients through function value-based estimates. There are different ZO optimizers for estimating the gradient. To better illustrate our framework, in this paper, we focus on the randomized gradient estimator (RGE) proposed by (Nesterov & Spokoiny, 2017), which approximates the FO gradient using finite differences of function values along randomly chosen direction vectors and has been used widely in the ZO optimization literature. Our idea can be applied to other ZO optimizers.

Given a loss function $L(\cdot)$ and a model with parameter $\theta \in \mathbb{R}^d$, the RGE employed by MeZO (Malladi et al., 2023), referred to as $\hat{\nabla}L(\theta)$, is to approximate $\nabla L(\theta)$ and is expressed using central difference:

$$\hat{\nabla}L(\theta) = gz \in \mathbb{R}^d, \tag{3}$$

$$g = \frac{L(\theta + \epsilon z) - L(\theta - \epsilon z)}{2\epsilon} \in \mathbb{R}^1, \tag{4}$$

where $z$ is a random direction vector drawn from the standard Gaussian distribution $\mathcal{N}(0, I)$, and $\epsilon > 0$ is a small perturbation step size, also known as the smoothing parameter. $g$ represents the **projected gradient** computed using the model's dual-forward passes. Notably, $g \in \mathbb{R}^1$ is just a scalar value and requires minimal memory space. The rationale behind RGE stems from the concept of the directional derivative (Duchi et al., 2015). As $\epsilon$ approaches 0, the directional derivative provides us an unbiased gradient estimator of $\nabla f(x)$. Thus, the RGE $\hat{\nabla}f(x)$ can be interpreted as an approximation of the FO gradient $\nabla f(x)$ using the directional derivative (Zhang et al., 2024). Zeroth-order stochastic gradient descent (ZO-SGD) follows a similar algorithmic framework to its first-order counterpart, SGD, but updates the parameters with the estimated gradient $\hat{\nabla}f(x)$ via zeroth order (function value) information for the descent direction. The parameters update of ZO-SGD is defined by:

$$\theta = \theta - \eta \cdot g \cdot z, \tag{5}$$

where $\eta$ is the learning rate. It is important to note that the variable $z$ in Equation 2 should be identical to the $z$ in Equation 1.

Specifically, the entire MeZO workflow is shown by Algorithm 3. The process initializes with parameters $\theta$ and iterates over a predetermined number of steps $T$. Each iteration samples a batch $B$ from dataset $D$ and employs a perturbation strategy. Parameters $\theta$ are

first perturbed positively to compute loss $\ell_+$, followed by a negative perturbation of $2\epsilon$ to compute $\ell_-$. Parameters are then reset to their original state for gradient estimation. The gradient is approximated by the difference $(\ell_+ - \ell_-)/(2\epsilon)$, representing the directional derivative along perturbed parameters. This projected gradient, combined with random Gaussian noise $z$, updates each parameter $\theta_i$, optimizing the loss function.

## D  More ZO2 Implementation Details

### D.1  Specific Dynamic Scheduler Design for Efficient Overlap

Figure 5a offers a schematic representation of the sequential, non-overlapping task execution within the basic ZO2 framework, particularly highlighting the inefficiencies of dual-forward passes without task overlap. Initially, input data is transferred from the CPU to the GPU, beginning with input processing through the embedding layer. Following this, each transformer block—ranging from Block 1 to Block n—undergoes a distinct cycle: upload to the GPU, execution of dual-forward computations, and subsequent offload back to the CPU upon completion of computations.

This linear processing sequence reveals a critical inefficiency: the GPU must idle while awaiting each block's offload back to the CPU, delaying the upload and processing of the subsequent block. This causes significant downtime for the GPU during offloads and for the CPU during uploads, as each must wait for the other's task completion before proceeding. The depicted lack of overlap between computation (green arrows) and communication (blue arrows) tasks pinpoints a crucial area for enhancement. Implementing an overlapped or asynchronous task management strategy, like Figure 5b, could markedly improve system efficiency and throughput, potentially reducing training times and optimizing the use of both CPU and GPU resources.

To overlap the data loading and computation process, we propose a dynamic scheduler, utilizing the asynchronous execution on different CUDA streams. Specifically, our scheduler includes three CUDA streams (Figure 1), which are utilized to control the $i$-th transformer block's computation, the $i + 1$-th block's uploading, and the $i - 1$-th block's offloading can occur concurrently. This design minimizes data transfer conflicts and maximizes GPU utilization by keeping computational and communication channels active.

However, designing this dynamic scheduler presents challenges when communication tasks outlast computation tasks, leading to potential errors. For example, if the upload of the $i$-th block is incomplete when its computation begins, this can lead to errors, as the GPU computes with an incomplete set of parameters. Similarly, if the computation of the $i$-th block is still ongoing when its offloading begins, it can also result in errors because the computation is disrupted by the removal of necessary data. An intuitive solution is to perform a global synchronization directly after initiating $U(W_{i+1})$, $C(W_i)$, and $O(W_{i-1})$ asynchronously. However, this approach is likely to introduce delays, or "bubbles," due to the varying execution speeds of the three tasks. To mitigate these issues, we establish two critical dependency relationships: (1) The offloading task of the $i$-th block is contingent upon the completion of its computation task, which, in turn, depends on the completion of its uploading task. (2) Each task must wait for the preceding task of the same type to complete; hence, the $i$-th task cannot commence until the $i - 1$-st task has concluded. To efficiently manage task execution, we use an asynchronous task launch sequence: $O(W_{i-1})$, $C(W_i)$, and $U(W_{i+1})$. Here, $O(W_{i-1})$ synchronously awaits the completion of both $C(W_{i-1})$ and $O(W_{i-2})$. Similarly, $C(W_i)$ will not start until both $U(W_i)$ and $C(W_{i-1})$ have finished. Lastly, $U(W_{i+1})$ must wait for the completion of $U(W_i)$, ensuring orderly and error-free task progression.

A significant advantage of our framework is that **the dual-forward mechanism doubles the computation time, while the communication time per block remains unchanged**. This enhancement substantially increases the likelihood of complete overlap between communication and computation tasks, especially since communication between the CPU and GPU is generally slower than computation on the GPU. Our following evaluations (Section 6) show that with ZO's unique dual-forward passes, which extend computation times compared

with the single forward pass, communication delays are no longer the primary bottleneck in most scenarios.

Moreover, special attention needs to be given to the embedding parameters and the LM head, as they represent the beginning and end of the model, respectively. By consistently maintaining both the embedding and LM head on the GPU, we circumvent the overhead linked to frequent transfers. For the embedding layer, simultaneous uploading of input data and embedding parameters could compete for interconnection bandwidth. Moreover, keeping the embedding layer on the GPU enables the pre-uploading of the first transformer block, effectively overlapping with the computations of the embedding layer. Meanwhile, continuously keeping the LM head on the GPU removes delays associated with its offloading—since no subsequent block computations overlap with this offloading—and facilitates weight sharing with the embedding layer, as noted in some conditions (Radford et al., 2019), thus consolidating related computations and enhancing operational efficiency. The detailed scheduler design to apply ZO2 on LLMs is shown in Algorithm 2.

### D.2 Efficient Memory Management via Reusable One Block Space on GPU

We adopt the strategy by (Ren et al., 2021) to optimize memory management by pre-allocating a reusable transformer block of memory on the GPU, eliminating the overhead of repeated memory allocations and deallocations during data transfers between the CPU and GPU. This memory is dynamically reassigned to each transformer block in sequence, speeding up data transfers and stabilizing GPU memory usage, thereby enhancing computational efficiency.

We also adopt the strategy outlined by Li et al. (2020), leveraging communication buckets to enhance the throughput of block communications. Specifically, we concatenate parameter fragments within blocks into contiguous memory buckets, thus improving communication efficiency.

### D.3 Efficient Parameter Update Strategy

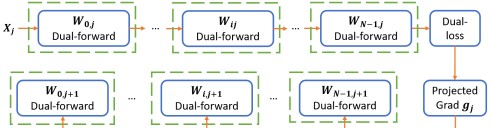
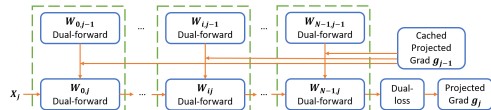

(a) Model parameter updates without the efficient strategy.

(b) Model parameter updates with the efficient strategy.

Figure 6: **Comparison of model parameters updates without/with efficient strategy**. (a) illustrates the process where, at the $j$-th iteration, the model computes the projected gradient $g_j$ using the dual-forward method and subsequently updates the model parameters. (b) demonstrates that at the $j$-th iteration, the model first updates the parameters using the previously saved projected gradient $g_{j-1}$, and then performs the dual-forward pass to compute the new projected gradient $g_j$.

In the ZO2 framework, the parameter update strategy is meticulously designed to precede the dual-forward computations of each transformer block. Traditionally, each transformer block is subjected to two distinct data transfer phases (Figure 6a, two green dotted boxes for each block): one for the dual-forward computations and another for applying gradient updates. This requirement stems from the fact that the (approximated) gradients are obtained only after completing the dual-forward computations for all blocks. For the first-order methods, the same offloading strategy requires parameters to be uploaded for the computation phase, offloaded upon completion, and then re-uploaded and offloaded again for the parameter update phase, given that the gradient dimensions match those of the parameters. This iterative process effectively doubles the communication load and extends the duration of training.

However, compared to the **interdependence** of dual-forward calculations across blocks, the parameter update process remains **independent** for each block, allowing us to reorder the operations. Once blocks are updated with the last iteration's gradients, only a single upload

and offload cycle is necessary for each block. This streamlined approach is only feasible in the ZO framework because, unlike first-order methods where the gradient dimensions are identical to those of the parameters, the projected gradient from ZO is only a scaler and can be persistently stored on the GPU. By implementing preemptive parameter updates, the framework significantly curtails the number of data transfers required per iteration (Figure 6b, one dotted box for each block). This adjustment not only halves the usage of interconnection bandwidth but also enhances the efficiency of the training process, thereby streamlining operations and reducing overhead.

### D.4 ZO2 in AMP Mode

Figure 4 illustrates the workflow of the ZO2 framework under AMP mode, which employs reduced precision formats to accelerate the training of LLMs. AMP leverages formats such as Tensor Float Point 32 (TF32), which provides higher computational throughput compared to Float Point 32 (FP32). AMP represents a compromise between FP32 and FP16, retaining the data storage format of 32 bits while offering computational speeds comparable to FP16. This adaptation allows for faster computation while maintaining the same communication speed as a purely FP32 workflow, which can complicate the computation-communication overlap. Therefore, a specialized framework must be designed to manage these unique challenges effectively. This acceleration is critical for enhancing training efficiency but introduces challenges in maintaining effective computation-communication overlap, as the data transfer still utilizes the FP32 format.

To address this, the ZO2 framework incorporates a compression mechanism where parameters are compressed to low-bit formats during offloading from GPU to CPU. This compression significantly reduces the data volume, enabling quicker transfers and mitigating bandwidth limitations. The current compression settings include bfloat16 and float16, which reduce the data size by 50%, and more aggressive reductions like float8, which compress to 25% of the original size.

Upon uploading these compressed parameters back to the GPU, they are decompressed and restored to FP32 for high-precision parameter updates. Subsequent computations, particularly the dual-forward passes, are then performed using the TF32 format to exploit the computational speed.

## E Codebase

```python
import torch
from torch.optim import AdamW
from transformers import OPTForCausalLM

# Model init
model = OPTForCausalLM.from_pretrained("facebook/opt-2.7b")
model.to("cuda")

# Optimizer init
optimizer = AdamW(model.parameters(), lr=1e-5)

# Training loop
for i in range(max_training_step):
    # Train
    training_input_ids, training_labels = ...   # get training data batch
    model.train()
    optimizer.zero_grad()
    output = model(input_ids=training_input_ids, labels=training_labels)
    output.loss.backward()
    optimizer.step()

    # Evaluate
    eval_input_ids, eval_labels = ...   # get eval data batch
    model.eval()
    with torch.no_grad():
        output = model(input_ids=eval_input_ids, labels=eval_labels)
```

(a) PyTorch first-order optimizer training.

```python
from zo2 import ZOConfig, zo_hf_init

# Model and optimizer init
zo_config = ZOConfig(method="mezo-sgd", zo2=True, offloading_device='cpu',
    working_device='cuda:0', lr=1e-5)
with zo_hf_init(zo_config):
    from transformers import OPTForCausalLM
    model = OPTForCausalLM.from_pretrained("facebook/opt-2.7b")
    model.zo_init(zo_config)

# Training loop
for i in range(max_training_step):
    # Train
    training_input_ids, training_labels = ...   # get training data batch
    model.zo_train()
    loss = model(input_ids=training_input_ids, labels=training_labels)

    # Evaluate
    eval_input_ids, eval_labels = ...   # get eval data batch
    model.zo_eval()
    output = model(input_ids=eval_input_ids, labels=eval_labels)

# Final training update
model.opt.zo_update(model)
```

(b) ZO2 training.

Figure 7: **Comparison of model training API with PyTorch first-order optimizer and with ZO2**.

ZO2 is encapsulated in approximately **5,500** lines of Python code, designed for ease of use, paralleling standard PyTorch training paradigms. The framework facilitates seamless integration and customization, enabling both researchers and practitioners to adapt it to diverse requirements efficiently. Figure 6 provides an illustrative example that demonstrates the API's similarity to conventional PyTorch usage, underscoring the user-friendly nature of ZO2.

## F  Full Experimental Setup

The experimental evaluation of our framework was conducted using the PyTorch deep learning library, integrated with NVIDIA CUDA streams to optimize parallel computation tasks. We selected the Open Pre-trained Transformer (OPT) (Zhang et al., 2022) model family (Table 4) as the subject of our experiments due to its open-source availability, widespread adoption in the research community, and diverse range of model sizes, ranging from 1.3 billion to 175 billion parameters, which allows for a comprehensive assessment of our framework's performance across varying scales of model complexity.

In our evaluation, MeZO (memory-efficient zerothorder optimizer) (Malladi et al., 2023) serves as the baseline method, as it is the most memory-throughput efficient ZO method currently. Our framework builds upon MeZO, reducing GPU memory usage while maintaining throughput and precision. All performance, including measurements of GPU memory usage and throughput, were conducted using the Stanford Sentiment Treebank (SST-2) Socher et al. (2013). The tests were performed on a system equipped with an NVIDIA A100 GPU with 80GB of memory and an AMD Milan CPU, operating under Python version 3.11, PyTorch 2.4.0, and CUDA 12.1. Our experiments further employed hyperparameters such as a learning rate of $1 \times 10^{-7}$, a batch size of 1, 100 training steps, and a sequence length of 2048, to rigorously test the framework under controlled conditions.

## G  More Experiments

### G.1  Evaluation of AMP Mode

Table 5: **Throughput (tokens/sec) results to validate AMP Mode.** AMP auto-cast with FP16 (top) and BF16 (below).

| Model | ZO2 (non-compress) | ZO2 (FP16) | ZO2 (BF16) | ZO2 (FP8) |
|---|---|---|---|---|
| OPT-1.3B | 4827 | 4770 (x0.988) | 4760 (x0.986) | 4802 (x0.995) |
| OPT-2.7B | 2811 | 2974 (x1.058) | 2974 (x1.058) | 2997 (x1.066) |
| OPT-6.7B | 1271 | 1641 (x1.291) | 1641 (x1.291) | 1662 (x1.308) |
| OPT-13B | 561 | 930 (x1.658) | 930 (x1.658) | 951 (x1.695) |
| OPT-30B | 286 | 416 (x1.455) | 416 (x1.455) | 425 (x1.486) |
| OPT-66B | 127 | 192 (x1.512) | 192 (x1.512) | 198 (x1.559) |
| OPT-175B | 43 | 65 (x1.512) | 65 (x1.512) | 68 (x1.584) |
| OPT-1.3B | 4565 | 4430 (x0.970) | 4430 (x0.970) | 4463 (x0.978) |
| OPT-2.7B | 2778 | 2816 (x1.014) | 2816 (x1.014) | 2818 (x1.014) |
| OPT-6.7B | 1273 | 1594 (x1.252) | 1594 (x1.252) | 1612 (x1.266) |
| OPT-13B | 678 | 910 (x1.342) | 910 (x1.342) | 924 (x1.363) |
| OPT-30B | 285 | 407 (x1.428) | 407 (x1.428) | 415 (x1.456) |
| OPT-66B | 127 | 188 (x1.480) | 188 (x1.480) | 194 (x1.528) |
| OPT-175B | 43 | 64 (x1.488) | 64 (x1.488) | 67 (x1.565) |

The efficiency of the AMP mode is shown in Table 5, where we evaluate the throughput using two AMP auto-cast computational data formats: FP16 and BF16. Additionally, we investigate the impact of various compression formats (FP16, BF16, and FP8) on communication and computation performance as detailed in Section D.4.

Across all models tested, a clear trend emerges: lower-bit compression formats consistently yield higher throughput. Notably, there is no significant difference in throughput between the 16-bit formats, FP16 and BF16, suggesting that the compression efficiency rather than the specific format type is the crucial factor in enhancing communication speed.

In most scenarios (specifically for the OPT models greater than 2.7B), employing low-bit compression results in superior throughput, underscoring the benefits of reducing data transfer volumes. However, exceptions are observed, such as with the OPT-1.3B model, where non-compressed data slightly outperforms the compressed formats. This outcome is attributed to the system being computation-bound rather than communication-bound. In such contexts, the additional computational demands imposed by the compression process do not sufficiently offset the benefits of reduced data transfer times, thereby introducing an overhead that detracts from the overall system efficiency.

### G.2 Analysis of More Experimental Settings

Table 6: **Different batch-size analysis.**

| Model | Batch-size | Memory Usage (MB) | | Throughput (tokens/sec) | |
|---|---|---|---|---|---|
| | | MeZO | ZO2 | MeZO | ZO2 |
| OPT-1.3B | | 8898 | 5098 (x0.57) | 1998 | 1955 (x0.97) |
| OPT-2.7B | 1 | 14514 | 5930 (x0.41) | 1104 | 1086 (x0.98) |
| OPT-6.7B | | 32930 | 8420 (x0.26) | 492 | 485 (x0.98) |
| OPT-13B | | 58762 | 10736 (x0.18) | 266 | 259 (x0.97) |
| OPT-1.3B | | 11733 | 8481 (x0.72) | 2126 | 2105 (x0.99) |
| OPT-2.7B | 2 | 18373 | 9312 (x0.50) | 1204 | 1193 (x0.99) |
| OPT-6.7B | | 34943 | 9683 (x0.27) | 561 | 556 (x0.99) |
| OPT-13B | | 61919 | 12315 (x0.19) | 297 | 294 (x0.99) |
| OPT-1.3B | | 16067 | 12887 (x0.80) | 2309 | 2296 (x0.99) |
| OPT-2.7B | 4 | 23731 | 14211 (x0.59) | 1268 | 1262 (x0.99) |
| OPT-6.7B | | 43683 | 17233 (x0.39) | 589 | 587 (x0.99) |
| OPT-13B | | 72159 | 19591 (x0.27) | 312 | 311 (x0.99) |
| OPT-1.3B | | 27529 | 22775 (x0.82) | 2358 | 2348 (x0.99) |
| OPT-2.7B | 8 | 37113 | 23639 (x0.63) | 1297 | 1294 (x0.99) |
| OPT-6.7B | | 57541 | 26207 (x0.45) | 604 | 602 (x0.99) |
| OPT-13B | | - | 31399 | - | 320 |

**Differential Batch-size and Sequence Length Analysis.** This analysis explores the impact of varying batch sizes and sequence lengths on the performance of the ZO2 compared to the MeZO baseline. Tables 6 and 7 present the memory usage and throughput metrics for different configurations of the OPT models, ranging from 1.3B to 13B parameters. Table 6 shows the results for different batch-sizes. As batch size increases, there is a consistent trend where ZO2 maintains throughput equivalency with MeZO across all model sizes, despite significant reductions in memory usage. Even at higher batch sizes, ZO2 demonstrates robust performance, showing almost no decrease in throughput relative to its MeZO counterpart. For example, in the OPT-1.3B model at a batch size of 8, the throughput remains constant at 2348 tokens/sec, maintaining operational efficiency irrespective of the increased computational load.

Table 7 illustrates the impact of sequence length on throughput. Similar to the batch-size analysis, increasing the sequence length does not compromise the throughput of ZO2, maintaining parity with the MeZO model across varying lengths. Notably, even at a sequence length of 8192 for the OPT-1.3B model, ZO2 sustains a throughput of 1279 tokens/sec, effectively handling larger input sizes without a drop in performance.

The analyses confirm that ZO2 effectively manages larger batch sizes and sequence lengths without sacrificing throughput. This resilience is crucial for practical deployments where varying input sizes and batch configurations are common, underscoring the scalability and robustness of the ZO2 approach in diverse operational environments.

Table 7: **Different sequence length analysis.**

| Model | Length | Memory Usage (MB) | | Throughput (tokens/sec) | |
|---|---|---|---|---|---|
| | | MeZO | ZO2 | MeZO | ZO2 |
| OPT-1.3B | | 8665 | 4471 (x0.51) | 1901 | 1830 (x0.96) |
| OPT-2.7B | 1024 | 14465 | 5389 (x0.37) | 1051 | 1013 (x0.96) |
| OPT-6.7B | | 31379 | 8405 (x0.26) | 439 | 426 (x0.97) |
| OPT-13B | | 56183 | 10717 (x0.19) | 244 | 243 (x0.99) |
| OPT-1.3B | | 8898 | 5098 (x0.57) | 1998 | 1955 (x0.97) |
| OPT-2.7B | 2048 | 14514 | 5930 (x0.41) | 1104 | 1086 (x0.98) |
| OPT-6.7B | | 32930 | 8420 (x0.26) | 492 | 485 (x0.98) |
| OPT-13B | | 58762 | 10736 (x0.18) | 266 | 259 (x0.97) |
| OPT-1.3B | | 11379 | 7581 (x0.67) | 1707 | 1692 (x0.99) |
| OPT-2.7B | 4096 | 19655 | 9023 (x0.45) | 1008 | 1001 (x0.99) |
| OPT-6.7B | | 38047 | 11763 (x0.30) | 509 | 505 (x0.99) |
| OPT-13B | | 64519 | 14915 (x0.23) | 275 | 272 (x0.99) |
| OPT-1.3B | | 32499 | 20507 (x0.63) | 1282 | 1279 (x0.99) |
| OPT-2.7B | 8192 | 38127 | 21351 (x0.55) | 786 | 784 (x0.99) |
| OPT-6.7B | | 54495 | 24115 (x0.44) | 446 | 445 (x0.99) |
| OPT-13B | | - | 30355 | - | 246 |

## G.3 Evaluation on Qwen3 Models

To evaluate the generalizability of ZO2 beyond the OPT family, we conduct experiments on Qwen3 (Yang et al., 2025). Compared to MeZO, ZO2 achieves significant GPU memory reduction with only a slight drop in throughput. This confirms that our framework remains highly efficient, even when applied to different LLM architectures. This further supports the adaptability of ZO2 across diverse model families.

Table 8: **Results of ZO2 performance for various Qwen3 models and FP16 modes.** The values in parentheses (x) represent the ratio of each measurement compared to the baseline MeZO (first column) configuration.

| Model | GPU Memory Usage (MB) ↓ | | Throughput (tokens/sec) ↑ | |
|---|---|---|---|---|
| | MeZO (FP16) | ZO2 (FP16) | MeZO (FP16) | ZO2 (FP16) |
| Qwen3-1.7B | 9329 | **8649(x0.92)** | **8320** | 6873(x0.82) |
| Qwen3-4B | 14357 | **9287(x0.64)** | **3955** | 3600(x0.91) |
| Qwen3-8B | 22995 | **10403(x0.45)** | **2391** | 1827(x0.76) |
| Qwen3-14B | 36877 | **13033(x0.35)** | **1421** | 1114(x0.78) |
| Qwen3-32B | 71195 | **13691(x0.19)** | **668** | 500(x0.74) |

