# OpenReview forum: "Scalable Zeroth-Order Fine-Tuning for Extremely Large Language Models with Limited GPU Memory"
_colmweb.org/COLM/2025/Conference — COLM 2025_

### Official Review · Reviewer_SBMa · 2025-05-06

**Rating:** 6
**Confidence:** 3
**Ethics Flag:** 1

**Summary:**

The paper proposes to shift some computation from GPU to CPU with LLM's zeroth-order fine-tuning. Specifically, it will load the transformer block one by one from CPU memory to GPU memory, and several tricks such as pipelining have been proposed to increase the throughput. Experimental results show that the proposed framework can handle larger models' fine-tuning while maintaining the same performance.

**Questions To Authors:**

1. Could you explain why the proposed framework works very well in FP16 but not well in FP32? It seems to me the proposed framework should be the same with different precisions.

**Reasons To Accept:**

1. The paper studies an important problem in reducing the GPU memory requirement for the large language model fine-tuning.
2. The experimental results demonstrate the effectiveness of the proposed framework.

**Reasons To Reject:**

1. The paper's organization and presentation need to be improved. Figure 1 is not helpful in showing the design of the proposed framework, and the figure that the paper refers to several times (Figure 5) is in the appendix. Two of the proposed tricks mentioned in the ablation study are in the appendix. Overall, I believe the paper can be much improved by prioritizing the content and details.
2. The proposed framework is only tested in the OPT model families. It would be better to show other models, although I believe the proposed framework should work in other model families.
3. I am not very familiar with the MLsys community, so I am not sure about the significance of the novelty in the paper. To me, the novelty is OK but not very significant.

---

> ### Author Response · Authors · 2025-06-02
>
> We thank Reviewer SBMa for their positive feedback and for acknowledging the importance of memory-efficient LLM fine-tuning.
>
> 1. On FP16 vs FP32 performance (reviewer's question):
>
> - We thank the reviewer for this insightful question and would like to clarify that our paper does not claim that the proposed framework performs poorly under FP32. The key performance comparisons are always made between ZO2 and MeZO under the same precision level.
>
> - We acknowledge that presenting FP32 and FP16 results in the same table (Table 2) may have unintentionally led to confusion. We apologize for this oversight. In the camera-ready version, we plan to split the results into separate tables for FP32 and FP16 to avoid this ambiguity.
>
> - Importantly, our textual discussion does not directly compare FP16 and FP32, because we believe such comparisons are less meaningful. Instead, we consistently focus on how ZO2 improves over MeZO under each respective precision configuration.
>
> - That said, if one insists on comparing FP32 and FP16, FP16 naturally has significant advantages in both memory usage (due to lower precision) and speed (due to Tensor Core acceleration).
>
> - Our framework supports both precisions and remains effective in either mode. We hope this clarifies the intended focus of our evaluation.
>
> 2. On applicability beyond the OPT family (point 2 of "Reasons To Reject"):
>
> We thank the reviewer for raising this point. While our main experiments focused on OPT for consistency with MeZO, ZO2 can be used for any decoder-only transformer.
>
> To demonstrate this, we conducted additional experiments on the Qwen3 model family—a recent, widely used decoder-only LLM series. Architecturally, models like LLaMA, Falcon, and Mistral share similar block structures and can likewise be partitioned for offloading, but we chose Qwen3 as a representative and up-to-date test case.
>
> As shown below, ZO2 consistently reduces peak GPU memory usage across Qwen3 models, with savings increasing at larger scales:
> | Model Size    | ZO Memory (MB) | ZO2 Memory (MB) | ZO2 / ZO (%) |
> |------|-------|---------|-------|
> | Qwen3-1.7B | 9329      | 8649     | 92%  |
> | Qwen3-4B    | 14357    | 9287     | 64%  |
> | Qwen3-8B    | 22995    | 10403   | 45%  |
> | Qwen3-14B  | 36877    | 13033   | 35%  |
> | Qwen3-32B  | 71195    | 13691   | 19%  |
> These results confirm that ZO2 is easily extensible to modern decoder-only LLMs, with excellent memory savings even for large-scale models.
>
> 3. On Figure 1 and Figure 5 (point 1 of "Reasons To Reject"):
> We thank the reviewer for this suggestion. In the camera-ready version, we will move Figure 5 into the main paper and revise or replace Figure 1 to better illustrate the system design.
>
> 4. On techniques mentioned only in the appendix (point 1 of "Reasons To Reject"):
> We agree and will move the key ablation details (e.g., parameter update and memory reuse) from the appendix into the main text in the camera-ready version.
>
> 5. On the significance of our novelty
> We appreciate the reviewer’s comment and would like to clarify that our work in fact introduces several complementary contributions across insight, method, and empirical validation. To better articulate the scope of these contributions, we summarize them below:
>
> - Insight-level contribution:
>   We observe that a core structural property of zeroth-order (ZO) optimization—its use of dual forward passes in the same direction—makes it fundamentally more compatible with CPU offloading than first-order (FO) training. Unlike FO methods, which require both forward and backward passes and thus transmit model parameters twice (once for each phase), ZO performs two forward passes using only a single parameter upload per block. Furthermore, because the two forward passes proceed sequentially in the same direction, ZO2 allows parameter communication to be effectively overlapped with computation. This structure enables our scheduler to be both simple and efficient, avoiding the complex interleaving logic typically required by FO offloading systems.
>
> - Method and system design:
>   (1) A lightweight dynamic scheduler using three CUDA streams that achieves high throughput with minimal control logic.
>   (2) An RNG state manager that preserves gradient correctness under block-wise, asynchronous execution—a requirement unique to our offloading setting.
>   (3) Several practical engineering optimizations, including block-local preallocated memory buffers, parameter update reordering to eliminate redundant transfers, and AMP-aware low-bit compression.
>
> - Empirical results:
>   (1) We enable fine-tuning of OPT-175B on a single 18GB GPU, to our knowledge, the first for ZO methods.
>   (2) ZO2 achieves up to 10× GPU memory savings, no accuracy loss, and comparable or improved throughput across model sizes (Tables 1–3).
>   (3) Our ablation study demonstrates that each proposed module—scheduler, memory reuse, update strategy—independently contributes to throughput gains.

---

### Official Review · Reviewer_Ka4C · 2025-05-08

**Rating:** 5
**Confidence:** 4
**Ethics Flag:** 1

**Summary:**

This paper introduces ZO2 with source code available. ZO2 implements an efficient ZO-offloading training framework on PyTorch with the following contributions:
(1) asynchronous operations with overlapping: we use 3 CUDA streams that are responsible for upload, compute, and offload. Each CUDA stream can have overlap, and their synchronization order is determined by the correctness of ZO (e.g. offload stream synchronizes with the compute stream, etc.)
(2) reusable memory space mapping
(3) RNG state manager and "fused" ZO updates with forward.

The evaluation mostly focus on the memory efficiency: MeZO on OPT-30B will cause CUDA OOM but ZO2 won't. ZO2 can effectively optimize OPT-175B with offloading without CUDA OOM.

**Questions To Authors:**

I don't have any other questions besides the "need clarification/modification" section in the weakness section.

**Reasons To Accept:**

The asynchronous computation and overlapping design in uploading/compute/offloading is unique to ZO and discussed sufficiently in the paper. This is a non-trivial system contribution. Figure 1, 4 & 5 is clear about this design.

**Reasons To Reject:**

My main concern is the lack of technical novelty on the system side. In my view, the ZO2's core contribution/technical challenge that worth  of discussion is only on the clever design of 3 CUDA stream and figuring out the right dependency and do overlapping (Figure 1, 4, `zo2/optimizer/mezo_sgd/zo2.py inner_zo_forward call`). The other techniques such as AMP during upload/offload, RNG manager, etc. is not worthy of discussion (although they are needed for correctness or efficiency purpose, the technique is either well-known or not hard to develop).

Need clarification / modification:
1.  "parameter updates are ingeniously fused with the dual forward passes" The code is implemented purely in PyTorch and in eager mode, the updated parameter will be read and written back to DRAM before the forward pass proceeds. **The use of the word "fused" is not accurate as it implies that we are performing *in-memory/online* update just before matmul or other operation in SRAM**. I would suggest to replace it with "combine/merge the call" unless the framework is actually doing fused *online* update before matmul/other ops in SRAM

2. "Reusable memory allocation to avoid costly CUDA malloc/free during block transfers" According to `zo2/optimizer/mezo_sgd/zo2.py, line 439-520`, we are just allocating a large contiguous tensor for upload and offload. This mainly saves the overhead time not necessarily the actual malloc/free time. **The "reusable" term is misleading as we are just allocating the large block right before upload/offload**. The same memory space won't be reused across layer and there is no custom memory allocator that pre-allocate a large memory pool and we reuse the pre-allocated space. *It is likely that we here only save the PyTorch overhead time instead of any actual NVidia CUDA malloc time.*

Minor but needed for more precise discussion on the system benefits:

As ZO2's main contribution is on the efficiency side with several system designs, a low-level trace (PyTorch profiler trace on CUDA stream is acceptable, ideally with detailed nsight trace) is definitely needed in the appendix or in the ablation study.

---

> ### Author Response · Authors · 2025-06-02
>
> We thank the reviewer for their detailed feedback and for recognizing the uniqueness of our asynchronous overlapping design using three CUDA streams (upload, compute, offload), as illustrated in Figures 1, 4, and 5 and implemented in the code. Below, we address the specific concerns and clarify the novel contributions of our work.
>
> 1. Clarifying our contributions (main concern):
> While we acknowledge that ZO2 does not introduce new optimization algorithms per se, we respectfully emphasize that the value of our work lies in co-designing an effective system architecture that exploits the structural advantages of ZO methods. We summarize our contributions below:
>
> - Insight-level contribution:
>   We observe that a core structural property of zeroth-order (ZO) optimization—its use of dual forward passes in the same direction—makes it fundamentally more compatible with CPU offloading than first-order (FO) training. Unlike FO methods, which require both forward and backward passes and thus transmit model parameters twice (once for each phase), ZO performs two forward passes using only a single parameter upload per block. Furthermore, because the two forward passes proceed sequentially in the same direction, ZO2 allows parameter communication to be effectively overlapped with computation. This structure enables our scheduler to be both simple and efficient, avoiding the complex interleaving logic typically required by FO offloading systems.
>
> - Method and system design:
>   (1) A lightweight dynamic scheduler using three CUDA streams that achieves high throughput with minimal control logic.
>   (2) An RNG state manager that preserves gradient correctness under block-wise, asynchronous execution—a requirement unique to our offloading setting.
>   (3) Several practical engineering optimizations, including block-local preallocated memory buffers, parameter update reordering to eliminate redundant transfers, and AMP-aware low-bit compression.
>
> - Empirical results:
>   (1) We enable fine-tuning of OPT-175B on a single 18GB GPU, to our knowledge, the first for ZO methods.
>   (2) ZO2 achieves up to 10× GPU memory savings, no accuracy loss, and comparable or improved throughput across model sizes (Tables 1–3).
>   (3) Our ablation study demonstrates that each proposed module—scheduler, memory reuse, update strategy—independently contributes to throughput gains.
>
> 2. On RNG state management (main concern):
> RNG state management is a necessary design element unique to ZO2’s offloading-aware execution model. MeZO applies Gaussian perturbations monolithically over the entire model, and thus requires only global seed control. In contrast, ZO2 performs block-wise CPU offloading, where each transformer block is independently uploaded, perturbed, dual-forwarded, and offloaded. Without precise RNG synchronization, the random perturbations used in parameter update and the dual forward passes would diverge across blocks, breaking the correctness of gradient estimation. Our RNG manager solves this by recording and restoring per-block RNG states, thereby ensuring correctness under asynchronous execution and enabling ZO2 to match MeZO’s accuracy.
>
> 3. On the absence of low-level profiling data ("minor but needed"):
> We agree that adding a low-level CUDA stream trace (e.g., from PyTorch Profiler or Nsight Systems) could enhance the presentation of our overlapping mechanism. We have collected such traces during development, which confirm the expected overlap between upload, compute, and offload streams. We have now made a representative Nsight Systems trace publicly available at the following anonymous link:
> [ https://anonymous.4open.science/r/zo2paper-D175 ]
> This trace confirms the correctness of our 3-stream overlap scheduling in practice.
> In the original submission, we chose not to include this trace due to space constraints, as we had already validated the effectiveness of our scheduler through the ablation study (Table 3). In that table, ZO2 achieves throughput nearly identical to MeZO, but removing the scheduler results in a significant throughput drop across all models. This clearly demonstrates the practical benefit of our overlap mechanism. We fully agree, however, that visualizing this via profiling is helpful and will include such analysis in the final version.
>
> 4. On the use of the term “fused” (point 1 of "need clarification / modification"):
> We thank the reviewer for this suggestion and agree that the term “fused” may cause confusion. We will revise the text in the camera-ready version to use “combined” or “scheduled together” to more accurately reflect our design.
>
> 5. On “reusable memory” (point 2 of "need clarification / modification"):
> We agree with the reviewer’s clarification and will update the terminology in the camera-ready version to refer to our strategy as a “preallocated block-local buffer” rather than “reusable memory,” to more precisely describe our implementation.

---

> > ### Comment · Reviewer_Ka4C · 2025-06-10
> > **Thank you for the reply**
> >
> > Thank you for the replies. The nsys profile shows the effectiveness of overlapping and high GPU utilization and I will raise my score accordingly.
> >
> > However, I still maintain my assessment for the novelty part. I understand RNG state management is required for correctness (as I said in the review), but this technique is not hard to develop (the RNG state management is an artifact of correct computation/communication dependency. Once we figure out the correct dependency, the RNG state is determined).
> >
> > I still believe the major contribution of this framework is only on the overlapping part and this alone does not sound sufficient.

---

> > > ### Author Response · Authors · 2025-06-11
> > >
> > > We sincerely thank the reviewer for raising their score and for acknowledging the effectiveness of our overlapping mechanism and the high GPU utilization observed in the nsys trace.
> > >
> > > While we respect the reviewer’s opinion regarding the perceived system novelty, we would like to clarify that the contributions of our work span three key dimensions: **(1) conceptual insight, (2) practical system design, and (3) experimental results**. We note that the reviewer’s comment appears to focus primarily on the second part (engineering implementation), while overlooking the insight-driven design motivation and the empirical achievements that are equally central to our contribution.
> > >
> > > (1) Insight-level contribution:
> > >
> > > Our work begins with a novel observation about the structural properties of zeroth-order (ZO) optimization: its use of two forward passes in the same direction makes it fundamentally more compatible with CPU offloading than first-order (FO) methods. In FO training, the forward and backward passes traverse the model in opposite directions, which necessitates transmitting parameters twice—once before the forward pass and again before the backward pass—if offloading is used. FO methods also require saving intermediate activations for gradient computation. In contrast, ZO performs two forward passes in the same direction using a single parameter upload per block and avoids activation storage entirely. Moreover, the two passes can be sequentially scheduled, enabling natural and effective overlap between communication and computation. This insight is foundational to our scheduling design and distinguishes ZO2 from FO-based offloading systems, which often require complex interleaving and incur higher communication costs.
> > >
> > > (2) System and engineering contributions:
> > >
> > > In system design, meaningful contributions often emerge from the integration of multiple components into a coherent and effective whole. Our implementation includes:
> > > - A dynamic 3-stream CUDA scheduler to enable asynchronous overlap of upload, compute, and offload;
> > >
> > > - A block-level RNG state manager to ensure correctness under pipelined and block-wise execution;
> > >
> > > - A memory preallocation strategy to reduce allocation overhead;
> > >
> > > - An efficient parameter update reordering strategy that eliminates redundant data transfers;
> > >
> > > - And support for AMP-style low-precision computation for further memory savings and throughput gains.
> > >
> > > These modules work together to enable stable and high-throughput execution under the asynchronous and fragmented pattern of offloading.
> > >
> > > (3) Experimental achievement:
> > >
> > > These system and engineering contributions collectively yield a practically significant result: we enable fine-tuning of OPT-175B on a single 18GB GPU, which, to our knowledge, is the first such result achieved by any zeroth-order training framework. This showcases both the practicality and efficiency of our approach. Our method achieves up to 10× GPU memory savings with little or no speed degradation across models. Moreover, experiments on the Qwen3 model family confirm ZO2’s applicability to modern decoder-only architectures beyond OPT, demonstrating its generality.
> > >
> > > In summary, while we understand the reviewer’s concerns about the novelty of individual engineering techniques, we believe that the insight, careful system integration, and strong empirical results make ZO2 a valuable contribution to memory-efficient LLM fine-tuning.

---

### Official Review · Reviewer_pm1a · 2025-05-13

**Rating:** 8
**Confidence:** 4
**Ethics Flag:** 1

**Summary:**

This paper presents a memory-efficient language model fine-tuning system by combining an existing zeroth-order optimization system with CPU offloading techniques. The authors first identified that zeroth-order optimization, due to its forward-pass-only nature, incurs less communication overhead when using CPU offloading. Then the authors introduced a series of technique: (1) the algorithm loads and offloads parameters in blocks, and the two forward passes and the parameter update from last iteration are done at the same time to avoid multiple communications; (2) the authors introduced a RNG (random number generator) manager to handle doing parameter update and forward passes at the same time; (3) the algorithm overlaps communication and computations; (4) reusable memory allocation to avoid malloc/free overhead; (5) support of low precision.

The experiment results are very strong: the algorithm performs the same as MeZO (a zeroth-order implementation) and saves up to 10x memory; there is very little speed degradation even given the heavy communication. The authors also present ablations to demonstrate the importance of each of the techniques above.

**Questions To Authors:**

Please see "reasons to reject".

**Reasons To Accept:**

The proposed ZO2 method is both innovative and very useful.

(1) The authors identified that zeroth-order optimization is more suitable for CPU offloading than first-order methods, as all the operations can be packed into one scan of the parameters and no activations need to be saved. This saves at least twice the amount of communication. The use of RNG manager such that the parameter update and forward passes can be done at the same time is also very cool.

(2) The authors put in a lot of engineering effort (scheduler, support low precision, reusable memory) to enable the high performance of the algorithm.

(3) The memory reduction is significant with almost no speed degradation, thanks to the above design. Wherever people use ZO and need further memory reduction, they should refer to this method.

**Reasons To Reject:**

The following is not a reason for rejection (and the authors do not need to add these for rebuttal), but just thoughts on the ZO method and experiments:

(1) Do we need more memory reduction? Considering the performance gap (and the fact that ZO still needs many more steps) between ZO and first-order optimization methods, I can imagine one of the main use cases of ZO is on-device update. Many of these mobile devices nowadays have unified memory (such as the M chips from apple), and hence "offloading" no longer makes a difference. It is of course out of the scope of this paper, but I think it would be great if the authors provide some discussions on this topic.

(2) How effective is ZO? Again, this is out of this paper's scope (which assumes ZO is effective and useful). I also understand that this paper follows the original MeZO paper's experiment setup. But it would be cool to see more experiments on other models (Llama for example) and more tasks (instruction tuning for example). Again, the authors do not need to conduct these for rebuttal and this is merely a suggestion.

---

> ### Author Response · Authors · 2025-06-02
>
> We thank the reviewer for their thoughtful and encouraging feedback, and for recognizing both the insight and system design behind ZO2, including the use of RNG, scheduler, and memory optimizations.
>
> We appreciate the recognition of ZO2’s insights and system design. As suggested, we have further validated the generality of ZO2 on additional models beyond OPT, specifically on the Qwen3 model family.
> The following table shows the peak GPU memory usage (in MB) of ZO vs. ZO2 across Qwen3 model sizes. ZO2 consistently outperforms the baseline, with greater memory savings as model size increases:
> | Model Size    | ZO Memory (MB) | ZO2 Memory (MB) | ZO2 / ZO (%) |
> |----------------|---------------------|-----------------------|--------------|
> | Qwen3-1.7B | 9329                     | 8649                       | 92%          |
> | Qwen3-4B    | 14357                   | 9287                       | 64%          |
> | Qwen3-8B    | 22995                   | 10403                     | 45%          |
> | Qwen3-14B  | 36877                   | 13033                     | 35%          |
> | Qwen3-32B  | 71195                   | 13691                     | 19%          |
> These results confirm that ZO2 is easily extensible to modern decoder-only LLMs, with excellent memory savings even for large-scale models.
>
> Regarding the comment on unified memory systems (e.g., Apple M-series chips), we focused our design and experiments on heterogeneous architectures with discrete CPU and GPU memory, as these remain widely used in large-scale LLM training across both academia and industry.
>
> That said, we very much appreciate the reviewer’s suggestion to consider unified memory systems, which indeed may reduce the need for explicit offloading due to their shared memory space. We agree this is a valuable direction for future work and may inspire further adaptation of ZO2-style techniques to such architectures.

---

> > ### Comment · Reviewer_pm1a · 2025-06-05
> > **Thank you**
> >
> > Thanks for the additional result! It's a strong paper and I will keep my current positive score.

---

> > > ### Author Response · Authors · 2025-06-11
> > >
> > > We sincerely thank the reviewer for their kind words and for maintaining a strong positive score. We truly appreciate your support and feedback.

---

### Decision · Program_Chairs · 2025-07-08

**Decision:**

Accept

**Comment:**

This paper introduces ZO2, a system enabling memory-efficient zeroth-order (ZO) fine-tuning of large language models—up to OPT-175B—on a single 18GB GPU. By leveraging the dual-forward structure of ZO optimization, ZO2 achieves efficient CPU-GPU offloading with minimal overhead. Key contributions include a 3-stream CUDA scheduler, RNG state management, and low-bit precision support.

Strengths:
- Addresses a highly practical challenge in LLM fine-tuning.
- Demonstrates strong empirical results with up to 10× memory savings and no accuracy loss.
- Well-integrated system design, with meaningful insights and thoughtful engineering.

Concerns:
- reviewer Ka4C noted that individual components (e.g., RNG manager, memory buffers) are not novel in isolation, though the integration and insight are recognized as valuable.

Two reviewers strongly supported acceptance (ratings of 8 and 6), emphasizing the method’s novelty and practical impact. A third reviewer, initially more critical (score 5), acknowledged the effectiveness of the system (especially the overlapping design) and raised their score after engaging with the authors' clarifications and data. Although concerns about the originality of some system components remain, the consensus supports that the work constitutes a valuable contribution.

Conclusion:
With general support from reviewers and effective rebuttals, this paper makes a timely and practical contribution. I recommend acceptance.